# Structural-Activity Relationship of Rare Ginsenosides from Red Ginseng in the Treatment of Alzheimer’s Disease

**DOI:** 10.3390/ijms24108625

**Published:** 2023-05-11

**Authors:** Xianwen Ye, Haixia Zhang, Qian Li, Hongmin Ren, Xinfang Xu, Xiangri Li

**Affiliations:** 1Centre of TCM Processing Research, Beijing University of Chinese Medicine, Beijing 102488, China; 20210941399@bucm.edu.cn (X.Y.); 20220935103@bucm.edu.cn (H.Z.); 20220935102@bucm.edu.cn (Q.L.); renhongmin0607@bucm.edu.cn (H.R.); 2Beijing Key Laboratory for Quality Evaluation of Chinese Materia Medica, Beijing University of Chinese Medicine, Beijing 102488, China; 3Institute of Regulatory Science for Traditional Chinese Medicine, Beijing University of Chinese Medicine, Beijing 102488, China

**Keywords:** red ginseng, rare ginsenosides, structure–activity relationship, Alzheimer’s disease, inflammasome

## Abstract

Rare ginsenosides are the major components of red ginseng. However, there has been little research into the relationship between the structure of ginsenosides and their anti-inflammatory activity. In this work, BV-2 cells induced by lipopolysaccharide (LPS) or nigericin, the anti-inflammatory activity of eight rare ginsenosides, and the target proteins expression of AD were compared. In addition, the Morris water maze test, HE staining, thioflavins staining, and urine metabonomics were used to evaluate the effect of Rh4 on AD mice. Our results showed that their configuration influences the anti-inflammatory activity of ginsenosides. Ginsenosides Rk1, Rg5, Rk3, and Rh4 have significant anti-inflammatory activity compared to ginsenosides *S*-Rh1, *R*-Rh1, *S*-Rg3, and *R*-Rg3. Ginsenosides *S*-Rh1 and *S*-Rg3 have more pronounced anti-inflammatory activity than ginsenosides *R*-Rh1 and *R*-Rg3, respectively. Furthermore, the two pairs of stereoisomeric ginsenosides can significantly reduce the level of NLRP3, caspase-1, and ASC in BV-2 cells. Interestingly, Rh4 can improve the learning ability of AD mice, improve cognitive impairment, reduce hippocampal neuronal apoptosis and Aβ deposition, and regulate AD-related pathways such as the tricarboxylic acid cycle and the sphingolipid metabolism. Our findings conclude that rare ginsenosides with a double bond have more anti-inflammatory activity than those without, and 20(*S*)-ginsenosides have more excellent anti-inflammatory activity than 20(*R*)-ginsenosides.

## 1. Introduction

Red ginseng is the principal product of ginseng, processed in steaming conditions. With modern technology development, the processing of red ginseng improved. Many red ginseng products are emerging at the same time, such as sun ginseng [1], black ginseng [2], fermented red ginseng [3], and puffed red ginseng [4]. Rare ginsenosides are nominated for being the least abundant constituents in ginseng. However, rare ginsenosides as activity components are abundant in red ginseng steamed at a higher temperature, such as ginsenosides Rg3, Rk1, Rg5, Rh1, Rk3, Rh4, et al. These rare ginsenosides have less polar and greater pharmacological activities, including anti-oxidant [5], anti-inflammatory [6], anti-cancer [7], and anti-apoptotic [8] activities.

Ginsenosides can incur structural change during red ginseng processing by demalonylation, decarboxylation, deglycosylation, and dehydration [9,10]. Ginsenoside is a triterpenoid saponin with a dammarane skeleton. Carbon-3, carbon-6, or carbon-20 combine with glycosyl residue by the ether bond. The 20(*S*)-ginsenosides are the natural products in ginseng. When fresh ginseng is processed into red ginseng, the deglycosylation in carbon-20 is easy. It is present in the 20(*S*) and 20(*R*) epimers, such as ginsenosides Rg3 and Rh1 (shown in Figure 1A,B). Furthermore, the dehydration in carbon-20 can present a double bond in carbon-20, 21 or carbon-20, 22, which offers positional isomers of the double bond at carbon-20(21) or carbon-20(22). While the double bond at carbon-20(21) has no present cis-trans isomers, such as ginsenosides Rk1 and Rk3 (shown in Figure 1C), the double bond at carbon-20(22) has cis-trans isomers, such as Rh4 and Rg5 (shown in Figure 1D). A previous study provided evidence that red ginseng, steamed at higher temperature and pressure, has the major components of these eight rare ginsenosides.

Alzheimer’s disease (AD) is neurodegenerative, with various pathophysiological aspects under investigation [11]. More and more experimental evidence, including genetic [12,13] and epigenetic [14] studies, transcriptome analysis of brains of patients with AD [15], and expression quantitative trait experiments in monocytes [16], supports a contributing role of innate immune mechanisms in AD. Microglia act as macrophages in the brain, which have the function of host defense and immune surveillance of the brain [17]. Microglia are known to lead to neuronal damage by releasing excessive pro-inflammatory cytokines and cytotoxic factors, which include nitric oxide (NO), tumor necrosis factor-α (TNF-α), interleukin-6 (IL-6), and interleukin-1β (IL-1β) [18]. Usually, microglia are activated for host defense in the brain. However, aberrant microglia activation can result in neuroinflammation, one of the leading causes of AD [19,20]. At the same time, pathological innate immune activation and subsequent production of inflammatory mediators, such as NLRP3 (NOD-, LRR- and pyrin domain-containing protein 3), ASC (apoptosis-associated speck-like protein), and caspase-1, promote the progress of AD and provide an effective therapeutic target for treating AD [21]. As we all know, many ginsenosides were reported to have an anti-inflammatory effect, especially rare ginsenosides such as Rh1, Rg3, Rk1, Rg5, Rk3, and Rh4. Nevertheless, there is little research to report the structure–activity relationship of anti-inflammatory activities between these ginsenosides. Additionally, whether rare ginsenosides can inhibit the expression of inflammatory mediator targets needs further investigation.

Here, we established an inflammatory cell model of macroglia induced by lipopolysaccharide (LPS) and/or nigericin. Griess reagents, Elisa, and Q-PCR determined the inflammatory factors released. This work compared the anti-inflammatory activity of ginsenoside (*S*-Rg3, *R*-Rg3, Rk1, Rg5, *S*-Rh1, *R*-Rh1, Rk3, and Rh4) and explored the structure–activity relationship between these ginsenosides. Further, we researched the rare ginsenosides’ activity in inhibiting the expression of AD-related target proteins, including NLRP3, caspase-1, and ASC. So, rare ginsenosides can be the candidate drugs for treating and preventing AD by inhibiting the expression of inflammatory mediator targets. This study verified the structure–activity relationship and effects on target proteins through molecular docking. The impact of Rh4 on improving the symptoms of AD mice was studied with an in vivo experiment. This study provides a reliable basis for preventing and treating AD with rare saponins.

## 2. Results

### 2.1. Protective Effect of Ginsenosides on LPS-Induced Microglial Inflammation

#### Ginsenosides for Inhibiting Inflammatory Factors Released

NO, TNF-α, and IL-6 are the primary inflammatory factors to assess tissue and cell inflammation. We further analyze the role of ginsenosides in the inflammatory factor release of BV-2 cells. Among the eight rare ginsenosides, ginsenosides Rk1 and Rg5 are the dehydration products of ginsenosides *S*-Rg3 and *R*-Rg3. Ginsenosides Rk3 and Rh4 are the dehydration products of ginsenosides *S*-Rh1 and *R*-Rh1. For selecting the adequate time of ginsenosides, ginsenoside Rg5 was treated at 0.5–4 h to determine the NO release in the supernatant. Figure 2A shows that ginsenoside Rg5 pre-treated for one h can significantly inhibit NO release, which is the better time for taking effect.

The activities of eight rare ginsenosides *S*-Rh1, *R*-Rh1, *S*-Rg3, *R*-Rg3, Rk1, Rg5, Rk3, and Rh4 for inhibiting NO, TNF-α, and IL-6 release are presented in Figure 2B–D. We can see that ginsenosides Rk1, Rg5, Rk3, and Rh4 have greater pharmacological activity than ginsenosides *S*-Rh1, *R*-Rh1, *S*-Rg3, and *R*-Rg3 in inhibiting NO, TNF-α, and IL-6 release, respectively. In addition, all ginsenosides Rk1, Rg5, Rk3, and Rh4 have dramatically reduced NO, TNF-α, and IL-6 release. However, all ginsenosides *S*-Rh1, *R*-Rh1, *S*-Rg3, and *R*-Rg3 can remarkably reduce NO. Ginsenosides *S*-Rh1, *R*-Rh1, and *S*-Rg3 can significantly reduce TNF-α. Only ginsenosides *S*-Rh1 and *S*-Rg3 can considerably reduce IL-6 levels. So, the double bond structure of ginsenosides exhibits greater pharmacological activity in anti-inflammation.

Similarly, ginsenosides *S*-Rh1 and *S*-Rg3 have a more significant effect than ginsenosides *R*-Rh1 and *R*-Rg3 in inhibiting NO, TNF-α, and IL-6 release, respectively. Ginsenoside *S*-Rh1 can significantly reduce the level of NO, TNF-α, and IL-6 release. However, ginsenoside *R*-Rh1 can significantly reduce NO and TNF-α release levels. Ginsenoside *S*-Rg3 can substantially reduce the level of NO, TNF-α, and IL-6 release. However, ginsenoside *R*-Rg3 can only significantly reduce the level of NO release. In addition, (*S*)-ginsenosides have a lower level of the same inflammatory factor than (*R*)-ginsenosides. So, we can indicate that the *S* epimers of ginsenosides have more excellent pharmacological activity than the *R* epimers of ginsenosides in inhibiting inflammatory factor release.

### 2.2. qRT-PCR Analysis of the Gene Expression of Inflammatory Factors

Ginsenosides can influence the release of inflammatory factors in the supernatant of BV-2 cells induced by LPS. Furthermore, total RNA from BV-2 cells was extracted to analyze how ginsenosides affect gene expression of TNF-α, IL-6, IL-1β, and iNOs. As shown in Figure 3, the LPS group showed a higher level of inflammatory factors (TNF-α, IL-6, IL-1β, iNOs), and ginsenosides Rk1, Rg5, Rk3, Rh4 can significantly reduce the expression of those. We can also see that ginsenosides *S*-Rh1, *R*-Rh1, *S*-Rg3, and *R*-Rg3 influenced gene expression of TNF-α, IL-6, IL-1β, iNOs in Figure 4. Ginsenosides *S*-Rh1, *R*-Rh1, *S*-Rg3, and *R*-Rg3 can dramatically reduce the gene expression of iNOs. Ginsenosides *S*-Rh1, *R*-Rh1, and *S*-Rg3 can markedly reduce the gene expression of TNF-α. Ginsenosides *S*-Rh1 and *S*-Rg3 can considerably reduce the gene expression of IL-6. Only ginsenoside *S*-Rh1 can drastically reduce the gene expression of IL-1β. Ginsenosides Rk1, Rg5, Rk3, and Rh4 have more significant pharmacological activity than ginsenosides *S*-Rh1, *R*-Rh1, *S*-Rg3, and *R*-Rg3 in anti-inflammation, which is coincident with inflammatory factor release.

In addition, ginsenosides *S*-Rh1 and *S*-Rg3 have a more significant effect than ginsenosides *R*-Rh1 and *R*-Rg3 in reducing the gene expression of TNF-α, IL-6, and IL-1β, respectively. Ginsenoside *S*-Rh1 can significantly reduce the levels of NO, TNF-α, IL-6, and IL-1β released. However, ginsenoside *R*-Rh1 can significantly reduce NO and TNF-α release levels. Ginsenoside *S*-Rg3 can considerably reduce the levels of NO, TNF-α, and IL-6 released. However, ginsenoside *R*-Rg3 can only significantly reduce the level of NO released. The *S* epimers of ginsenosides exhibit much stronger anti-inflammation activity than the *R* epimers of ginsenosides from the count of anti-inflammatory factors.

### 2.3. Molecular Docking Analysis for S and R Epimers of Ginsenosides

In Elisa and qPCR experiments, *S* and *R* epimers of ginsenosides have different anti-inflammation activities. Molecular docking was conducted to research the interaction relationship further. Appendix A shows the detailed parameters of ginsenosides *S*-Rh1, *R*-Rh1, *S*-Rg3, and *R*-Rg3 docking with target proteins of iNOs, IL-6, TNF-α (4NOS, 1ALU, 2AZ5). The molecular docking mode and interactions of 2AZ5 are shown in Figure 5. We can see the bond energy of *S*-Rh1, *R*-Rh1, *S*-Rg3, and *R*-Rg3 was less than −5.3 kcal/mol, which indicates the stability of the complex. The *S* epimers of ginsenosides have many hydrogen bonds and higher scores than the *R* epimers of ginsenosides. This illuminates why the *S* epimers of ginsenosides have greater anti-inflammation than the *R* epimers of ginsenosides at the molecular level.

### 2.4. qRT-PCR Analysis of the Protein Expression of AD

More and more reports reveal that the cause and development of AD are related to brain inflammation. The proteins NLRP3, ASC, and caspase-1 are vital in preventing and treating AD. So, we established an inflammatory model with LPS and nigericin, an activator of NLRP3. Once the NLRP3 was activated, ASC and caspase-1 influenced the following and produced IL-1β. Compared to the anti-inflammation of eight rare ginsenosides, we can see that Rk1, Rg5, Rk3, and Rh4 have a more significant effect than ginsenosides *S*-Rh1, *R*-Rh1, *S*-Rg3, and *R*-Rg3.

So, ginsenosides Rk1, Rg5, Rk3, and Rh4 were utilized to study the relationship with these target proteins of AD. From Figure 6, we can see that ginsenosides Rk1, Rg5, Rk3, and Rh4 can significantly reduce the IL-1β level of BV-2 supernatant, which indicates a high correlation with target proteins NLRP3, ASC, and caspase-1. Furthermore, RT-PCR was applied to analyze the gene expression of these target proteins. We can see that ginsenosides Rk1, Rg5, Rk3, and Rh4 can significantly reduce the gene expression of NLRP3, ASC, and caspase-1 in BV-2.

### 2.5. Laser Confocal Microscope Analysis of the Protein Expression of AD

Furthermore, ginsenoside Rh4 was used to analyze the influence on the proteins NLRP3 and ASC with a Laser confocal microscope. We can see NLRP3 with red fluorescence was expressed in both the control and model groups. The model group had significantly more expression than the control group. Ginsenoside Rh4 can dramatically reduce the protein expression of NLRP3 in BV-2 (Figure 7). Similarly, ASC with green fluorescence was expressed in the control and model groups. The model group had significantly more expression than the control group. Ginsenoside Rh4 can markedly reduce the protein expression of ASC in BV-2 (Figure 8).

Therefore, these ginsenosides may play an essential role in inflammasome signaling by inhibiting the expression of NLRP3, ASC, and caspase-1. The NLRP3 inflammasomes promote AD’s progression and provide a therapeutic target for AD. So, these results can elucidate the mechanism of ginsenosides in preventing and treating AD.

### 2.6. Molecular Docking Analysis for Ginsenosides to the Potential Proteins of AD

Molecular docking was conducted to research the interaction relationship between ginsenosides Rk1, Rg5, Rk3, and Rh4 and AD potential proteins. Appendix A shows the detailed parameters of ginsenosides Rk1, Rg5, Rk3, and Rh4 docking with target proteins of NLRP3, ASC, and caspase-1 (2ANQ, 6KI0, and 5MTK). The molecular docking mode and interactions of 5MTK are shown in Figure 9. We can see that the bond energy of ginsenosides Rk1, Rg5, Rk3, and Rh4 was less than −5.8 kcal/mol, which indicates the stability of the complex and the great anti-inflammation of ginsenosides.

### 2.7. Rh4 Ameliorates Dementia Conditions in APP/PS1 Mice

#### Effect of Rh4 on APP/PS1 Mouse Behavior

Water maze experiments can effectively evaluate the learning and memory ability of mice. The time of crossing the platform is an important indicator of the space search experiment. At a certain time, the number of times the experimental animals crossed the original platform was recorded. The more times the animal crossed the original platform, the better its spatial learning and memory ability [22]. Figure 10A shows the behavioral trajectories of different groups of mice. Based on the experimental results of the above cell experiments, we found that ginsenoside Rh4 had good activity, so we carried out in vivo experiments. It can be seen that there is a significant difference between the model group and the blank group, and the times of the model group are significantly less than that of the blank group. The positive drug donepezil hydrochloride group and ginsenoside Rh4 group showed substantially increased numbers of AD mice crossing the platform, and showed improvements in AD mouse behavior and memory disorders, indicating that ginsenoside Rh4 has pharmacological activity in the prevention and treatment of AD.

Figure 10B,C shows the escape latencies and target quadrant dwell times of the different groups of mice. Latency is an important indicator of the positioning and navigation stage of the Morris water maze, and it is the time required for the animal to successfully find the platform for the first time after each entry into the water. Its length also represents the quality of the animal’s spatial learning and memory ability, and a short incubation period indicates that the animal’s learning and memory ability is good [23]. The time in the quadrant where the platform is located is an index to evaluate the learning ability of animals. The longer the time and distance of animals in this quadrant, the better the spatial memory ability of animals [24]. It can be seen that the escape latency of the model group is significantly higher than that of the blank group. The positive drug donepezil hydrochloride group and the ginsenoside Rh4 group can dramatically reduce the escape latency of the mice. At the same time, the target quadrant residence time of the model group was significantly less than that of the blank group. The positive drug donepezil hydrochloride group and ginsenoside Rh4 group could significantly improve AD mouse target quadrant residence time.

### 2.8. HE Observed Pathological Changes in the Mouse Brain

HE staining showed that compared with the control group, the hippocampal CA1 and CA3 regions of mice in the model group were significantly different. The cells in the hippocampal CA1 region of mice in the model group were loosely arranged. In contrast, the cells in the hippocampal CA1 area of mice in the ginsenoside Rh4 group were significantly improved and tightly arranged (Figure 11A–C). There were significant differences in the hippocampal CA3 region of the model group, with cell solidification atrophy and hyper staining of the whole cell. The ginsenoside Rh4 group had an improved effect on cells in the hippocampal CA3 region of mice (Figure 11D–F). Ginsenoside Rh4 has a significant protective effect on hippocampal neuronal cells in AD mice.

### 2.9. Th-S Staining to Observe the Changes of Aβ in the Mouse Brain

The fluorescent dye thioflavin s can specifically bind to mature Aβ amyloid protein and has green fluorescence, which can be used to detect Aβ amyloid deposition in AD mice. The staining of the cerebral cortex of mice in different treatment groups is shown in Figure 11G–I. As can be seen from the figure, there was a pronounced deposition of amyloid beta in the cerebral cortex of mice in the model group compared with the control group. The ginsenoside Rh4 group can significantly reduce the amount and area of amyloid beta deposition in the cerebral cortex of mice. Ginsenoside Rh4 affects amyloid beta deposition in the cerebral cortex of AD mice.

### 2.10. Rh4 Regulates Multiple Metabolic Pathways in AD Mice and Plays a Therapeutic Role

After methodological investigation, the results showed good repeatability and high stability (Appendix A, Appendix A. Multivariate statistical analysis showed that the control group, model group, and Rh4 group could be significantly separated, indicating that the urine metabolism of AD mice changed significantly compared with the normal group. Administration of Rh4 can regulate urine metabolism, indicating that there are some mechanisms to alleviate metabolic disorders (Figure 12A,B).

The characteristic ions of two groups of samples can be obtained with partial least square analysis (Figure 12C,D). The molecular formula of the compound can be calculated according to the exact molecular weight of the characteristic ions. Then the structure of the compound can be preliminarily determined by searching the database and consulting the literature. The structure of the compound is further determined according to the fragment information, and the attribution of characteristic ions is shown in Table 1. They are the characteristic ions of urine samples in positive and negative ion modes, including 11 in positive and 13 in negative ion modes.

### 2.11. Analysis of Metabolic Pathway in Urine Samples

The metabolic pathway information of potential biomarkers in related databases such as METLIN, HMDB, and KEGG can be presented visually in graphs and tables. The results showed that 24 potential biomarkers were involved in six metabolic pathways (Figure 12E), including the TCA cycle, the glyoxylate and dicarboxylate metabolism, the primary bile acid biosynthesis, the phenylalanine metabolism, the sphingolipid metabolism, and the tryptophan metabolism. Among them, the TCA cycle, the glyoxylate and dicarboxylate metabolism, and the sphingolipid metabolism have relatively large impact values. The primary bile acid biosynthesis can also be significantly enriched.

## 3. Discussion

As we all know, ginsenosides are the effective compounds of ginseng products. Ginsenosides are named “Rx,” where the “R” stands for the root, and the “x” describes the chromatographic polarity in alphabetical order [25]. Customarily, the seven fingerprint ginsenosides (Rb1, Rb2, Rc, Rd, Re, Rg1, and Rf) are often measured to standardize ginseng extracts. Except for ginsenoside Rf, the other six ginsenosides are the most abundant ginsenosides in *Panax ginseng*, which are called major ginsenosides [26]. Other ginsenosides are called rare ginsenosides in ginseng products, except for these major ginsenosides. Previous research has indicated that ginseng steaming at a high temperature can transform major ginsenosides into rare ones. Ginsenosides *S*-Rg3, *R*-Rg3, Rk1, Rg5, *S*-Rh1, *R*-Rh1, Rk3, and Rh4 are the main components of ginseng steamed at a high temperature, which is much higher than in traditional red ginseng [9].

Pharmacological activities were directly related to the structure of ginsenoside. The 20(*S*)-malonyl-ginsenosides are an initial form of ginsenosides in fresh ginseng. Interestingly, in red ginseng processing, the length of the sugar moiety in ginsenoside was reduced, and stereoisomers occurred at C-20, such as *R*, *S*-epimers, and double-bond isomers. Figure 13 shows the formation of *S* and *R* epimers of ginsenoside Rg3 in red ginseng processing. Ginsenoside Rd is a protopanaxadiol (PPD) type of ginsenoside and precursor of ginsenoside Rg3, which has a glycosyl at chiral C-20. Deglycosylation produces a carbocation, which is then attacked by hydroxide anion from both sides with equal probability. As a result, ginsenoside transforms into retention configuration and inversion configuration. Figure 14 is shown the formation of double-bond positional and cis-trans isomers of ginsenosides in red ginseng processing. When the chiral C-20 of ginsenoside Rg3 is further dehydrated, the double bond occurs at carbon-20(21) and carbon-20(22). The chiral C-20 connects a methylene group. The double bond at this site has no cis-trans isomers and is transformed into ginsenoside Rh4. However, the double bond at carbon-20(22) has a pair of cis-trans isomers for the rotation of the C-C single bond, i.e., ginsenoside Rg5 and Rz1. Similarly, ginsenoside Rg1 is a protopanaxatriol (PPT) type of ginsenoside and the precursor of ginsenoside Rh1, the epimers (*S* and *R*) of ginsenoside Rh1 and ginsenoside Rk1, Rh4 are produced by deglycosylating and dehydrating, respectively.

In terms of anti-cancer activity, 20(*S*)-ginsenosides have a more substantial effect than their 20(*R*)-stereoisomer, and ginsenosides with a double bond at C-20(21) exhibit more activities than those at C-20(22) [27]. In other pharmacological activities, 20(*R*)-Rg3 shows higher adjuvant effects on the OVA-induced immune system than 20(*S*)-Rg3 in mice [28], 20(*S*)-Rg3 epimer exhibited higher pharmacological effects in insulin secretion and AMPK activation than 20(*R*)-Rg3 [29]. Ginsenosides Rk1 and Rg5, which have a double bond at C-20, have great pharmacological activities in anti-cancer [30], anti-inflammation [31], and anti-erectile dysfunction [32]. However, there is little research to analyze the anti-inflammatory activity of ginsenosides in different stereoisomers. So, in this work, we focused on eight rare ginsenosides to research their structure–activity relationship for anti-inflammatory activity. The *S* epimers of ginsenoside have greater anti-inflammation than the *R* epimers of ginsenoside by inhibiting the inflammatory factors’ release and expression. Furthermore, the double bond of ginsenoside has greater anti-inflammation than the *S* and *R* epimers of ginsenoside by inhibiting the inflammatory factors’ release and expression.

The ginsenoside stereoisomers may exhibit a different effect on the same pharmacological activity, and those results are not contradictory. The experimental design of other animal models and cell lines may lead to different conclusions based on diverse therapeutic targets. Thus, further research should be performed to elucidate the underlying mechanisms of these stereoisomers. In addition, the interaction of rare ginsenosides with AD target proteins needs to be further studied.

Microglia, the resident immune macrophages in the central nervous system (CNS), play a crucial role in responding to brain injuries and infections [33]. Microglia can be stimulated to an M1 phenotype by LPS or IFN-γ for the expression of pro-inflammatory cytokines such as TNF-α, IL-1β, IL-6, superoxide, NO, or M2 phenotype by IL-4/IL-13 for resolution of inflammation and tissue repair [34]. More and more evidence indicates that microglia-mediated neuroinflammation contributes significantly to the pathogenesis of neurodegenerative diseases, such as AD, Parkinson’s disease (PD), amyotrophic lateral sclerosis (ALS), and Huntington’s disease (HD) [35].

NLRP3 inflammasomes comprise NLRP3, ASC, and pro-caspase1 (precursor caspase-1), which activates caspase-1 to release IL-1β and IL-18 [36]. The activation of NLRP3 inflammasomes enhances Aβ aggregation. It drives tau hyperphosphorylation, which promotes the progression of NFTs (Neurofibrillary Tangles) and SPs (Senile Plaques) in AD [21,37]. Once the NLRP3 inflammasome is activated, it triggers ASC helical fibrillar assembly. It then makes ASC fibrils into large para-nuclear ASC specks [38,39]. In vitro, ASC speck formation and release can be induced in pre-stimulated mouse microglia or human THP-1 cells with exposure to NLRP3 inflammasome activators such as nigericin and ATP [21,40].

There is pervasive metabolic dysregulation in vivo with AD-related pathological changes. The tricarboxylic acid cycle was shared between aging and AD [41]. In this experiment, the contents of cis-aconitic acid and isocitrate in the urine of AD mice in the model group were significantly lower than those in the blank group, which indicated that AD mice had abnormal energy metabolism. The glyoxylate and dicarboxylate metabolism is related to the significant decrease in glycine concentration in PScDKO mice. Although glycine is the simplest amino acid, it plays an essential role in the pathogenesis of AD [42]. Bile acids (BAs) have primary regulatory and signaling functions and seem dysregulated in AD. The primary BA of serum concentrations in AD was significantly lower compared to cognitively normal older adults, indicating that bile acid metabolism plays a vital role in AD development [41]. AD was reported to be closely linked with abnormal lipid metabolism (such as glycerolipids, glycerophospholipids, and sphingolipids) and phenylalanine metabolism. The experiment showed that the contents of C20 sphingosine were significantly lower than those in the blank group but upregulated phenylalanine metabolism in the hippocampus, indicating sphingolipid metabolites and phenylalanine metabolism can be used as biomarkers for AD diagnosis [42,43,44]. Activating tryptophan (Trp) metabolism along the Kyn pathway can prevent excessive inflammation and induce long-term immune tolerance. With an increase in age and age-related diseases, the levels of Trp and Kyn in the whole body change. In addition, regulating Trp metabolism can aggravate or prevent inflammation-related diseases. Studies have shown that low tryptophan levels can lead to weight loss, emotional disorders, and cognitive impairment [45]. The above literature indicates that there may be disorders of energy metabolism, lipid metabolism, and inflammation-related pathways in AD, and Rh4 plays a reverse regulatory role in these metabolic pathways, thus exerting the therapeutic effect on AD.

Accounts of ginsenosides’ great anti-inflammation activity play an essential role in preventing and treating AD by inflammasome signaling, shown in Figure 15. In this signaling, LPS and TNF-α are used for activating the formation of NLRP3. Meanwhile, nigericin is used for activating the aggregation of NLRP3. In general, LPS and nigericin are both used to stimulate inflammasome signaling at the same time. Ginsenosides Rk1, Rg5, Rk3, and Rh4, in particular, can inhibit the formation and aggregation of NLRP3 by suppressing the release of inflammatory factors and the gene expression of ASC and caspase-1. That is the molecular mechanism of ginsenosides in preventing and treating AD.

## 4. Materials and Methods

### 4.1. Antibodies and Reagents

The standards of ginsenosides *S*-Rg3 (ST03850120MG) and *R*-Rg3 (ST02530120MG) were purchased from Shanghai Nature Standard Biotechnology Co., Ltd. (Shanghai, China). The standards of ginsenosides *S*-Rh1 (PS010067), *R*-Rh1 (PS010068), Rk1 (PS010828), Rg5 (PS180126-03), Rk3 (PS010059), and Rh4 (PS100051) were purchased from Chengdu Push Bio-Technology Co., Ltd. (Chengdu, China). The purity of these standards was more than 98%, stored in a refrigerator under 4 °C. Other reagents related to cell culture were purchased from Corning (One Riverfront Plaza Corning, New York, NY, USA). LPS (*Escherichia coli* serotype 055: B5, L2880) was acquired from Sigma Chemical Co. (St. Louis, MI, USA). Nigericin (S01D8L49263, purity ≥ 97%) was obtained from Shanghai Yuanye Bio-Technology Co., Ltd. (Shanghai, China). A nitric oxide assay kit (DEM106) was bought from Beijing BioDee Biotechnology Co., Ltd. (Beijing, China). Elisa kits of TNF-α (KE10002), IL-6 (KE10007), and IL-1β (KE10003) were purchased from Proteintech Group Inc. (Chicago, IL, USA).

### 4.2. Cell Experiment

#### 4.2.1. Cell Culture and Treatment

Microglia cell line BV-2 originated from murine and was purchased from the Cell Resource Center of Peking Union Medical College (Beijing, China). Then the cell was cultured in DMEM (10-013-CVR, Mediatech Inc., New York, NY, USA) supplemented with 10% FBS (900-108, Gemini, CA, USA), 100 U/mL penicillin (30-002-Cl, Mediatech Inc., New York, NY, USA), and 100 μg/mL streptomycin (30-002-Cl, Mediatech Inc., New York, NY, USA) at 37 °C in a humidified atmosphere of 5% CO_2_. The stock solutions of ginsenosides were prepared in DMSO, the final concentration of which in the cell culture media was less than 0.1%. LPS was used at a concentration of 1 μg/mL for microglia stimulation. LPS-primed microglia were also activated with nigericin (10 μM) to assess the release of ASC, NLRP3.

#### 4.2.2. Biochemical and Elisa Analysis

Microglia cells (1 × 10^5^ cells per well in a 24-well plate) were pre-treated for 1 h with 50 μM ginsenosides *S*-Rh1, *R*-Rh1, *S*-Rg3, *R*-Rg3, Rk1, Rg5, Rk3, Rh4 and nigericin for 6 h, and then stimulated for 24 h with LPS (1 μg/mL). The supernatants of different cultured microglia were collected after LPS stimulation for 24 h, the NO concentrations determined with a nitric oxide assay kit, and the Elisa kit, respectively, determined the TNF-α, IL-1β, and IL-6 concentrations.

### 4.3. Quantitative PCR and Immunocytochemistry (ASC, NLRP3) in BV-2 Cell

Total RNA was extracted using TRIzol Reagent (Life Technologies Corporation, Carlsbad, CA, USA) following the manufacturer’s instructions. RNA concentration and integrity were analyzed using a Nanodrop OneC microvolume UV-Vis spectrophotometer and agarose gel electrophoresis. RNA was reverse transcribed to single-stranded cDNA using the TIANScriptⅡRT kit (KR106, Tiangen Biotech, Beijing, China). Relative mRNA levels were quantified using TransStart Top Green qPCR SuperMix (Transgen Biotech, Beijing, China). The reaction conditions were 10 min at 95 °C, followed by 40 cycles of 15 s at 95 °C, 60 s at 60 °C, and then melt curve analysis to identify PCR specificity. The sequences of the primers used for qPCR were listed as follows in Table 2.

Nonspecific staining was performed with 5% goat serum for 1–2 h. The cells were incubated at 4 °C overnight with primary antibody anti-ASC (67824S, 1:500, Rabbit mAb, CST) and anti-NLRP3 (MAB7578, 5 μg/mL, Monoclonal Rat IgG_2A_, R&D). Goat Anti-rabbit IgG(H + L)-Alexa Flexa^®^488 (DE0635) and Goat Anti-rat IgG(H + L)-Dylight 594 (A23440) were used as secondary antibodies at a dilution of 1:250 and 1:500. The cells were sequentially stained for 15 min with Hoechst 33342. Fluorescence staining was visualized on a laser confocal microscope.

### 4.4. Molecular Docking Analysis

The three-dimensional (3D) coordinates of target proteins for iNOs, IL-6, TNF-α, NLRP3, ASC, and caspase-1 were downloaded from the Protein Data Bank (http://www.rcsb.org, accessed on 26 October 2022) and the PDB IDs were 4NOS, 1ALU, 2AZ5, 2ANQ, 6KI0, and 5MTK, respectively. The crystal structure of these proteins was determined with X-ray diffraction and the resolution was less than 3 Å. The 3D structures of eight ginsenosides were downloaded from ChemSpider (http://www.chemspider.com, accessed on 26 October 2022) and drawn using the ChemBio3D Ultra 14.0 software. Molecular docking studies were performed to investigate the binding mode of ginsenosides to target proteins by using Autodock 4.2.6 and Discovery Studio 4.5. The docking results of receptors and ligands were visually analyzed using the PyMOL 2.0.6 software.

### 4.5. Animal Experiment

#### Animal Grouping and Drug Administration

Constituting around 70% of individuals with AD, women have a greater lifetime risk for AD than men, and display approximately a threefold higher rate of disease progression with a broader spectrum of cognitive symptoms [46,47]. Therefore, only female animals were used in this experiment. APP/PS1 double transgenic mice were purchased from Beijing Hufukang Biotechnology Co., Ltd. (Beijing, China), five months old, weighing about 20 g, female. The temperature of the animal feeding room is 24–26 °C, the light is 14 h, the dark is ten h, and the water and food are free.

The animal experiment was reviewed and approved by the Laboratory Animal Ethics Review Committee of the Beijing University of Chinese Medicine (BUCM-4-2019060501-2037). Thirty APP/PS1 double transgenic mice were randomly divided into three groups, ten mice in each group, namely, model group, ginsenoside Rh4 group (dose of 20 mg/kg), and donepezil hydrochloride positive drug group (dose of 2 mg/kg). The model group and control group (ten mice of C57BL/6J strain) were given the same volume of distilled water instead for four months.

### 4.6. Morris Water Maze Test, HE Staining, and Thioflavin S Staining

Four months after drug administration, the mice first completed the water maze experiment, including the navigation experiment and space exploration experiment, to evaluate the learning and memory of mice, as described previously [47]. The diameter of the circular pool was 120 cm, the height was 50 cm, the water depth in the pool was 30 cm, the bottom of the pool was white, and the water temperature was kept at (23 ± 2) °C. The pool was divided into four quadrants, and a quadrant was selected for placement of a platform equidistant from the pool wall and pool center. The platform was 12 cm in diameter, 29 cm in height, and submerged 1 cm in water. In the experimental training stage, the mice were placed in the maze for training once a day and adapted to the environment. After six consecutive days, learning and memory function tests were performed. An automatic camera and Xmaze analysis system were used to track and record animal movement trajectories in real-time.

At the end of the administration, the mice were placed in a metabolic cage, and 12 h urine was collected from each group and stored in a refrigerator at −80 °C. At the end of the experiment, all mice were anesthetized with 1% pentobarbital (65 μL/10 g, i.p.). The mice were sacrificed, and the brain tissues were stored in formalin solution, dehydrated in gradient ethanol, transparent in xylene, immersed in wax, embedded in paraffin, sectioned, and stained with HE after xylene dewaxing. The histopathological changes were observed under a light microscope. Brain tissue was stained with thioflavins and photographed using a confocal laser.

### 4.7. Metabolomics Study of Urine

The urine samples were removed from the −80 °C refrigerator and thawed at room temperature. After high-speed centrifugation (14,000 rpm, 10 min, 4°C), the supernatant was vortexed and mixed with an equal volume of ultrapure water. The filtrate was passed through a 0.22 μm microporous filter membrane. All the details about metabolic analysis are contained in the Appendix A documentation.

### 4.8. Statistical Analysis

All data are shown as the mean ± SD of triplicate samples. The significance between the control and treated groups was examined with two-tailed Student’s *t*-tests or a one-way ANOVA test. Any *p*-values of less than 0.05 were considered significant.

## 5. Conclusions

In conclusion, the anti-inflammatory activity of eight rare ginsenosides in red ginseng was compared using comprehensive in vitro data, which indicated that the configuration of ginsenosides was closely connected with anti-inflammatory activity. It was proven that Rh4 has to regulate and improve its effect on AD mice through in vivo experiments. Rare ginsenosides with a double bond have more significant anti-inflammatory activity. The *S* configuration of ginsenosides has more excellent anti-inflammatory activity than the *R* configuration. Ginsenosides Rk1, Rg5, Rk3, and Rh4 can significantly reduce the target proteins NLRP3, ASC, and caspase-1 of AD in BV-2 cells by inhibiting inflammasome signaling. The tricarboxylic acid cycle, glyoxylate and dicarboxylate metabolism, and the sphingolipid metabolism play an important role in preventing and treating AD with Rh4. Red ginseng, abundant in these ginsenosides, can be a valuable medicament for preventing and treating AD.

## Figures and Tables

**Figure 1 ijms-24-08625-f001:**
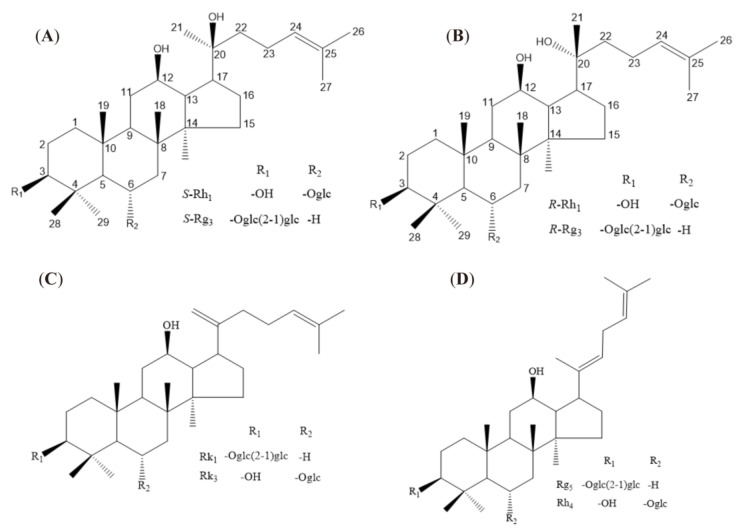
Structure of eight rare ginsenosides. The numbers in (**A**,**B**) represent the positions of carbon atoms, and R1 and R2 in the figure represent substituents (**C**,**D**).

**Figure 2 ijms-24-08625-f002:**
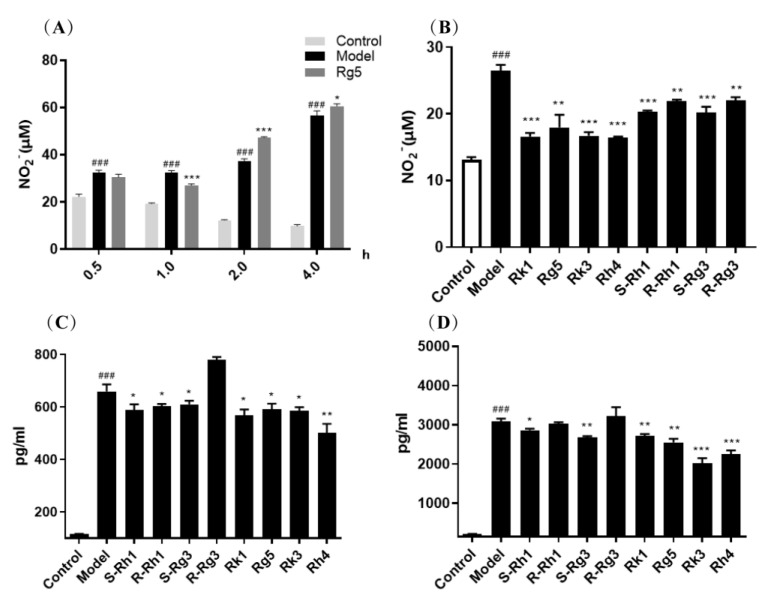
Eight rare ginsenosides affect the release of NO, TNF−α, and IL−6 in LPS−induced microglia cells. (**A**): Pre−treated ginsenoside Rg5 at different times; (**B**): NO release inhibition; (**C**): TNF−α release inhibition; (**D**): IL−6 release inhibition. The bar represents the mean ± standard error of the mean (*n* = 3). The Student’s *t*-test determined significant differences (### *p* < 0.001 compared with normal cells) or ANOVA (* *p* < 0.05, ** *p* < 0.01, *** *p* < 0.001 compared with LPS−treated cells).

**Figure 3 ijms-24-08625-f003:**
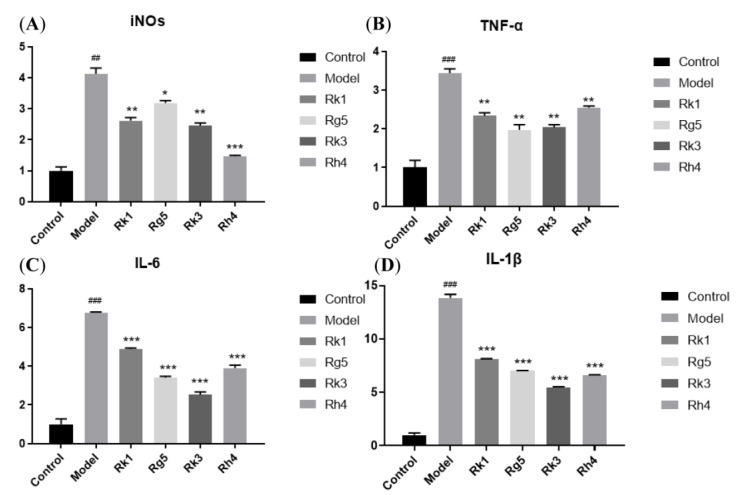
Ginsenosides Rk1, Rg5, Rk3, and Rh4 affect gene expression of inflammatory factors in LPS-induced microglia cells. (**A**): iNOs mRNA expression; (**B**): TNF-α mRNA expression; (**C**): IL-6 mRNA expression; (**D**): IL-1β mRNA expression. The bar represents the mean ± standard error of the mean (*n* = 3). The Student’s *t*-test determined significant differences (## *p* < 0.01, ### *p* < 0.001 compared with normal cells) or ANOVA (* *p* < 0.05, ** *p* < 0.01, *** *p* < 0.001 compared with LPS-treated cells).

**Figure 4 ijms-24-08625-f004:**
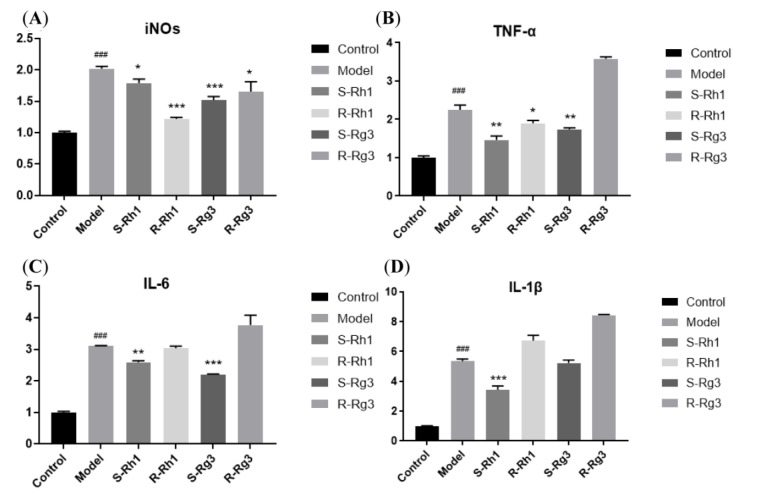
Ginsenosides *S*-Rh1, *R*-Rh1, *S*-Rg3, and *R*-Rg3 affect the expression of inflammatory factors in LPS-induced microglia cells. (**A**): iNOs mRNA expression; (**B**): TNF-α mRNA expression; (**C**): IL-6 mRNA expression; (**D**): IL-1β mRNA expression. The bar represents the mean ± standard error of the mean (*n* = 3). The Student’s *t*-test determined significant differences (### *p* < 0.001 compared with normal cells) or ANOVA (* *p* < 0.05, ** *p* < 0.01, *** *p* < 0.001 compared with LPS-treated cells).

**Figure 5 ijms-24-08625-f005:**
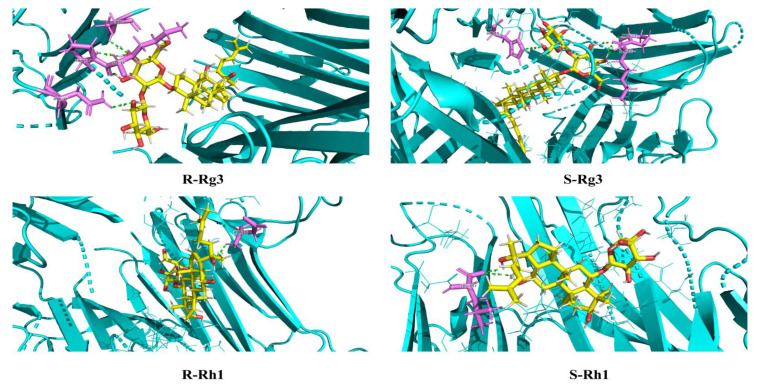
Molecular docking mode and interactions between 2AZ5 and ginsenosides *S*-Rh1, *R*-Rh1, *S*-Rg3, *R*-Rg3.

**Figure 6 ijms-24-08625-f006:**
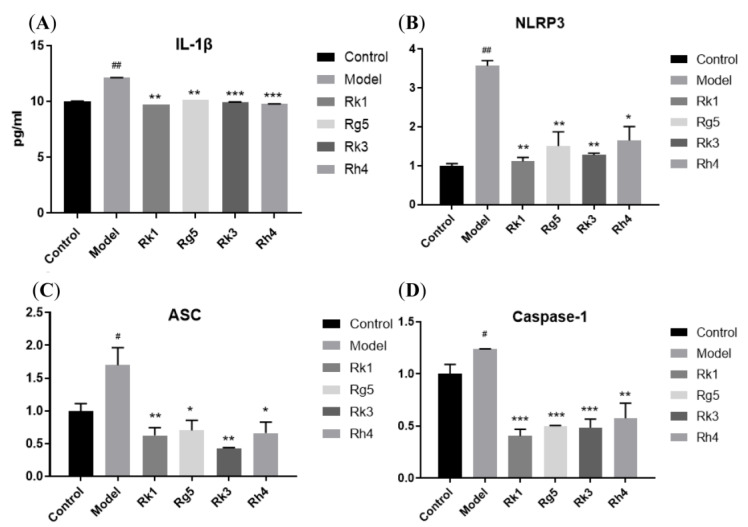
Ginsenosides Rk1, Rg5, Rk3, and Rh4 affect IL-β and inflammatory proteins in LPS and nigericin-induced microglia cells. (**A**): IL-1β release in supernatant; (**B**): NLRP3 mRNA expression; (**C**): ASC mRNA expression; (**D**): caspase-1 mRNA expression. The bar represents the mean ± standard error of the mean (*n* = 3). The Student’s *t*-test determined significant differences (# *p* < 0.05, ## *p* < 0.01 compared with normal cells) or ANOVA (* *p* < 0.05, ** *p* < 0.01, *** *p* < 0.001 compared with LPS-treated cells).

**Figure 7 ijms-24-08625-f007:**
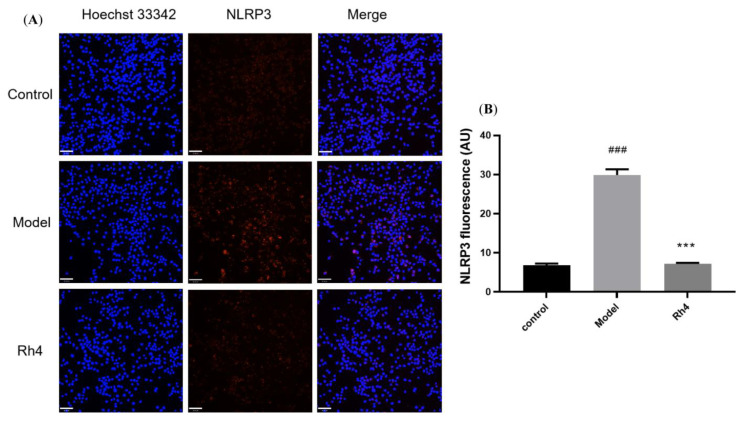
Effect of ginsenoside Rh4 on NLRP3 expression in LPS and nigericin-induced microglia cells. (**A**): NLRP3 (red) and nuclei (blue) using a laser confocal microscope; (**B**): Quantitative analysis of NLRP3 fluorescence intensity. Scale bars, 70 μm. The bar represents the mean ± standard error of the mean (*n* = 3). The Student’s *t*-test determined significant differences (### *p* < 0.001 compared with normal cells, *** *p* < 0.001 with LPS and nigericin-treated cells).

**Figure 8 ijms-24-08625-f008:**
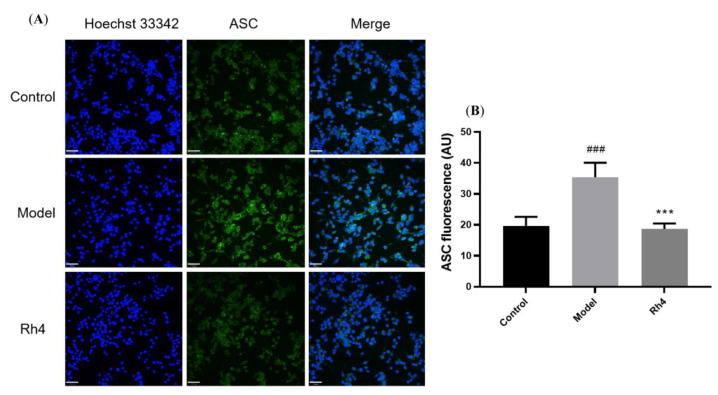
Ginsenoside Rh4 effects on ASC expression in LPS and nigericin-induced microglia cells. (**A**): ASC (green) and nuclei (blue) using a laser confocal microscope; (**B**): Quantitative analysis of ASC fluorescence intensity. Scale bars, 70 μm. The bar represents the mean ± standard error of the mean (*n* = 3). The Student’s *t*-test determined significant differences (### *p* < 0.001 compared with normal cells, *** *p* < 0.001 with LPS and nigericin-treated cells).

**Figure 9 ijms-24-08625-f009:**
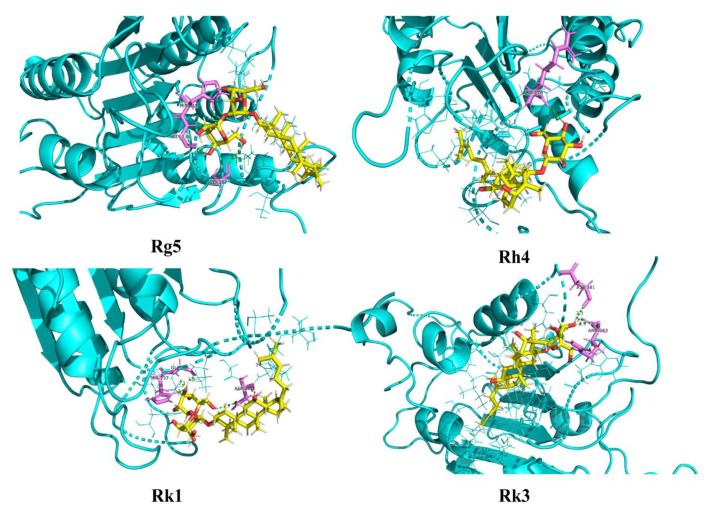
Molecular docking mode and interactions between 5MTK and ginsenosides Rk1, Rg5, Rk3, and Rh4.

**Figure 10 ijms-24-08625-f010:**
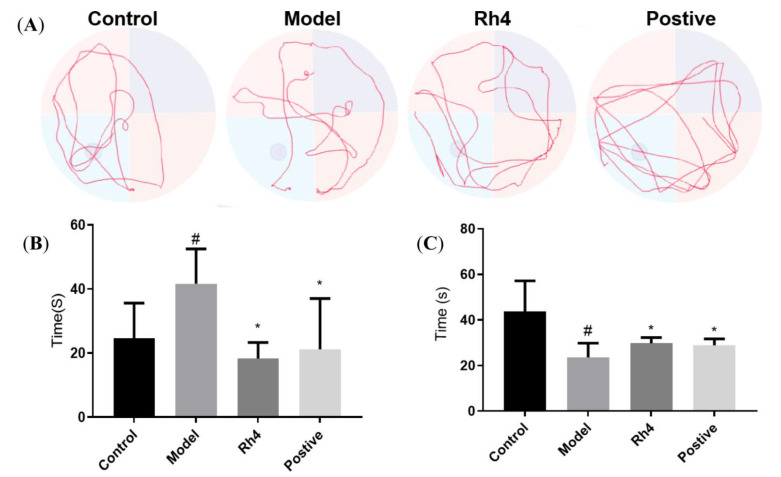
Mouse water maze, escape latency, and target quadrant stays time. (**A**), Water maze behavior trajectory; (**B**,**C**), Comparison of escape latency and target quadrant residence time of mice treated with different groups. (*n* = 6; #: compared with the control group, # *p* < 0.05; * compared with the model group, * *p* < 0.05).

**Figure 11 ijms-24-08625-f011:**
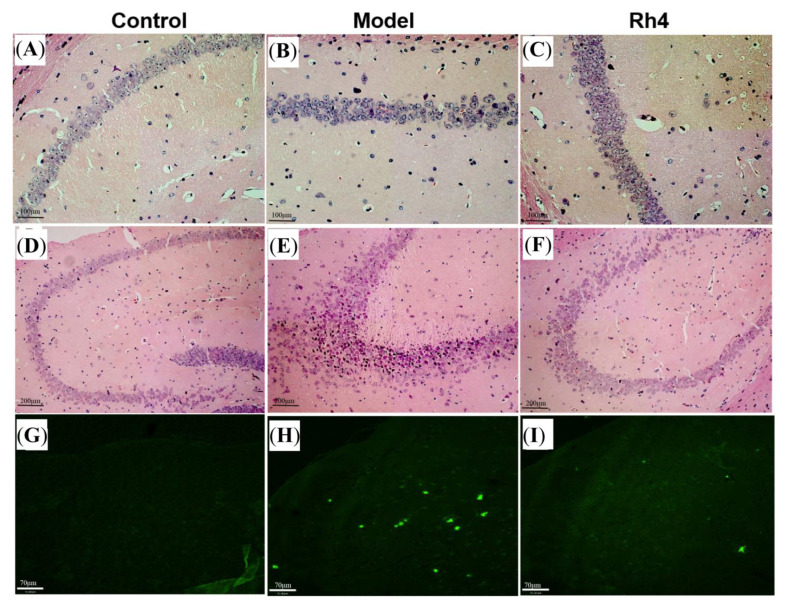
HE staining in hippocampal CA1 and CA3 regions and A β plaque staining in the cerebral cortex of mice in different treatment groups; (**A**–**C**), HE staining of CA1 region, 20×; (**D**–**F**), HE staining of CA3 region, 10×; (**G**–**I**), cortical Aβ plaque staining.

**Figure 12 ijms-24-08625-f012:**
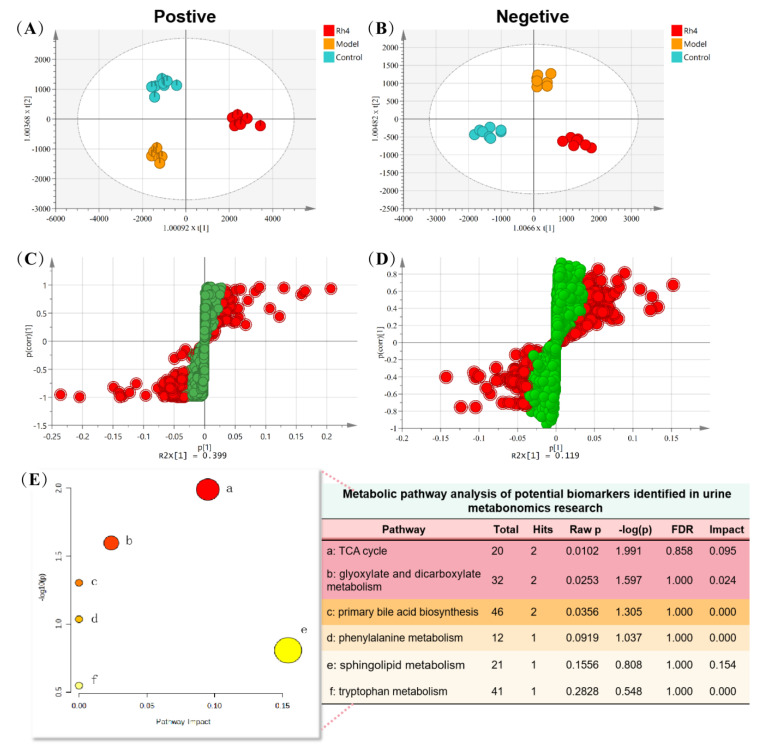
Multivariate statistical analysis of urine metabolism and enrichment of metabolic pathway in different groups of mice. (**A**,**B**), OPLS−DA analysis; (**C**,**D**), S−plot analysis, red represents components with significant differences between the two groups; (**E**), metabolic pathway analysis. a: TCA cycle, b: glyoxylate and dicarboxylate metabolism, c: primary bile acid biosynthesis, d: phenylalanine metabolism, e: sphingolipid metabolism, f: tryptophan metabolism. Red represents significant pathways.

**Figure 13 ijms-24-08625-f013:**
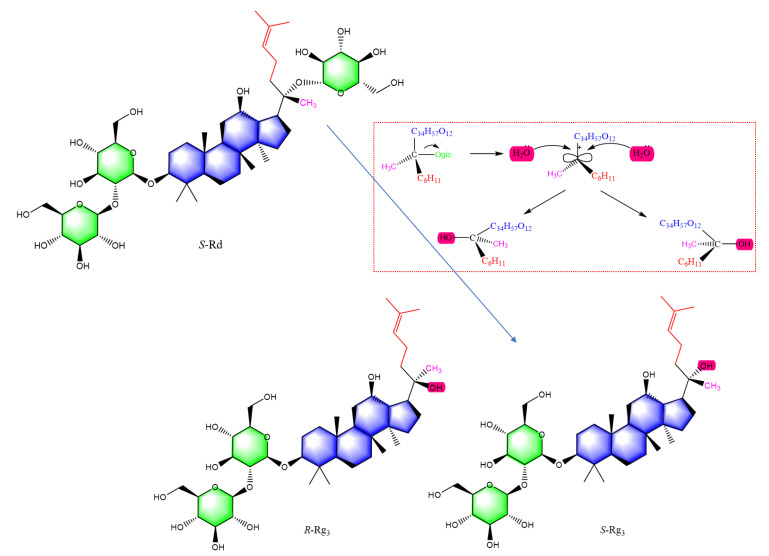
The formation of *S* and *R* epimers of ginsenoside Rg3 in red ginseng processing.

**Figure 14 ijms-24-08625-f014:**
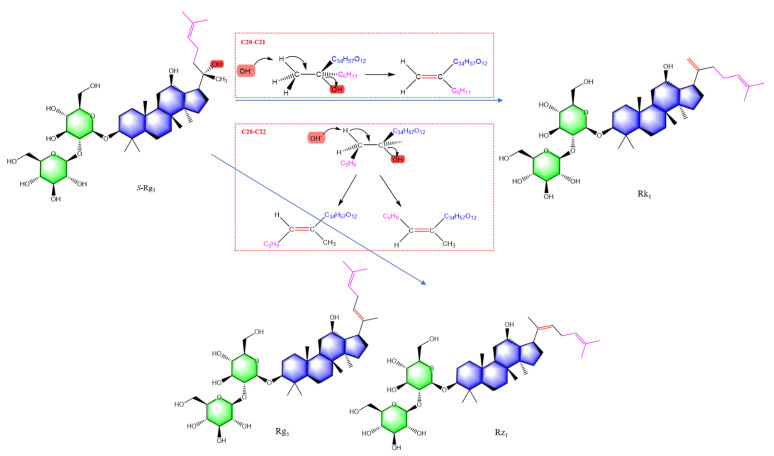
The formation of double bond positional and cis-trans isomers of ginsenosides in red ginseng processing.

**Figure 15 ijms-24-08625-f015:**
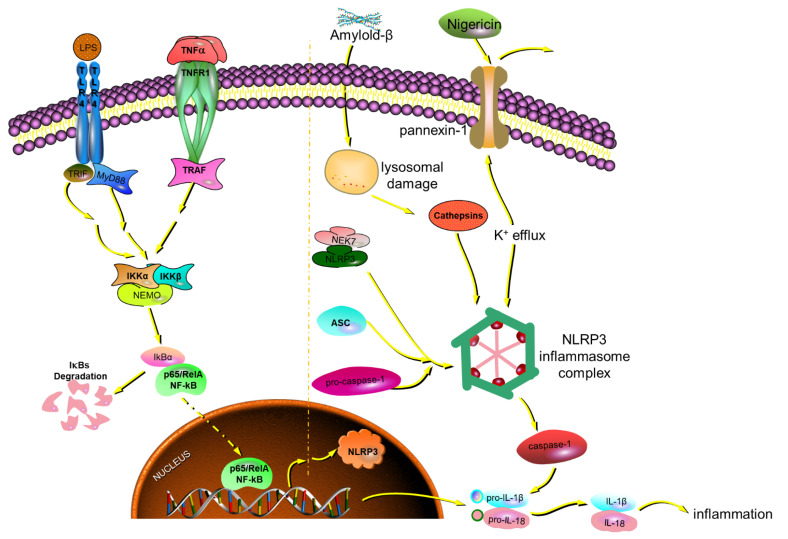
The mechanism of rare ginsenosides of red ginseng in preventing and treating AD.

**Table 1 ijms-24-08625-t001:** Characteristic ions of urine samples in positive and negative ion mode.

No.	t_R_ (min)	Precursor Ions and/or Adduct Ions	Exact Mass	Error (ppm)	Formula	Identification
1	0.86	173.0076 [M − H]^−^	173.0086	−5.78	C_6_H_6_O_6_	cis-Aconitate acid
2	0.92	191.0193 [M − H]^−^	191.0192	0.52	C_6_H_8_O_7_	Isocitric acid
3	0.98	254.9802 [M − H]^−^	254.9811	−3.53	C_6_H_8_O_9_S	Ascorbic acid-2-sulfate
4	2.66	286.0014 [M + FA-H]^−^	286.0021	−2.62	C_9_H_7_NO_5_S	Indole-3-carboxilic acid-O-sulphate
5	2.81	158.0809 [M − H]^−^	158.0817	−5.06	C_7_H_13_NO_3_	Valerylglycine
6	3.18	206.0445 [M − H]^−^	206.0453	−3.88	C_10_H_9_NO_4_	4-(2-Aminophenyl)-2,4-dioxobutanoic acid
7	3.19	178.0498 [M − H]^−^	178.0504	−3.37	C_9_H_9_NO_3_	Hippuric acid
8	3.31	212.0017 [M − H]^−^	212.0018	−0.47	C_8_H_7_NO_4_S	Indoxyl sulfate
9	3.49	222.0788 [M − H]^−^	222.0767	9.46	C_11_H_13_NO_4_	N-Acetyl-L-tyrosine
10	3.68	283.0818 [M − H]^−^	283.0818	0.00	C_13_H_16_O_7_	p-Cresol glucuronide
11	4.91	377.1959 [M − H]^−^	377.1964	−1.33	C_21_H_30_O_6_	18-Hydroxycortisol
12	8.59	407.2787 [M − H]^−^	407.2798	−2.70	C_24_H_40_O_5_	Cholic acid
13	10.26	391.2840 [M − H]^−^	391.2849	−2.30	C_24_H_40_O_4_	Chenodeoxycholic acid
14	2.21	206.0460 [M + H]^+^	206.0453	3.40	C_10_H_7_NO_4_	Xanthurenic acid
15	2.83	162.0560 [M + H]^+^	162.0555	3.09	C_9_H_7_NO_2_	Indole-3-carboxylic acid
16	2.84	338.0883 [M + H]^+^	338.0876	2.07	C_15_H_15_NO_8_	3-Indole carboxylic acid glucuronide
17	2.84	338.0883 [M + H]^+^	338.0876	2.07	C_15_H_15_NO_8_	2,8-Dihydroxyquinoline-beta-D-glucuronide
18	3.20	164.0705 [M + H]^+^	164.0711	−3.66	C_9_H_9_NO_2_	3-Methyldioxyindole
19	3.81	162.0554 [M + H]^+^	162.0555	−0.62	C_9_H_7_NO_2_	2-Indolecarboxylic acid
20	4.90	379.2115 [M + H]^+^	379.2121	−1.48	C_21_H_30_O_6_	18-Hydroxycortisol
21	6.32	181.0870 [M + H]^+^	181.0864	3.31	C_10_H_12_O_3_	3-Methoxybenzenepropanoic acid
22	10.46	302.3061 [M + H]^+^	302.3059	0.66	C_18_H_39_NO_2_	Sphinganine
23	10.86	328.3220 [M + H]^+^	328.3215	1.52	C_20_H_41_NO_2_	D-erythro-C20-Sphingosine
24	11.89	330.3385 [M + H]^+^	330.3372	3.94	C_20_H_43_NO_2_	eicosasphinganine

**Table 2 ijms-24-08625-t002:** PCR primer sets for RT-PCR analysis.

Gene	Forward (5′→3′)	Reverse (3′→5′)	Length
iNOs	GAGCGAGTTGTGGATTGTC	CCAGGAAGTAGGTGAGGG	133 bp
TNF-α	GTGAAGGGAATGGGTGTT	GGTCACTGTCCCAGCATC	198 bp
IL-6	CCACCAAGAACGATAGTCAA	TTTCCACGATTTCCCAGA	392 bp
IL-1β	TGGGCTGGACTGTTTCTA	ATCAGAGGCAAGGAGGAA	184 bp
NLRP3	ACCTCCAAGACCACTACGG	CAGCCAGTGAACAGAGCC	118 bp
ASC	TGCCAGGGTCACAGAAGT	CCAGGTCCATCACCAAGTA	209 bp
Caspase-1	CCCCAGGCAAGCCAAATC	TGAGGGTCCCAGTCAGTCC	202 bp
GAPDH	ATGTACGTAGCCATCCAGGC	AGGAAGGAAGGCTGGAAGAG	420 bp

## Data Availability

The datasets analyzed during the current study are available from the corresponding author on reasonable request.

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
