# Peer review of "Structural-Activity Relationship of Rare Ginsenosides from Red Ginseng in the Treatment of Alzheimer’s Disease"

_ijms, 2023, doi:10.3390/ijms24108625_

Round 1
Reviewer 1 Report
In this study, author investigated the relationship between structure of ginsenosides and their anti-inflammatory activity. Author showed that configuration of ginsenosides influences their anti-inflammatory activity. Additionally, author found one of the Ginsenoside Rh4 improve the cognitive function in Female AD mouse.
Overall manuscript is well-written and clearly presents the main findings of the study. Here are some suggestions and corrective comments to enhance the quality of manuscript:
1) In materials and methods, it is better to add the catalog number of every products or kits mentioned in the method section along with the company.
2) In animal experiment, Is there any specific reason of using only Female transgenic mice rather than using both male and female mice to avoid the gender bias. The results may not be representative of the entire population and may not be applicable to male mice or humans. Better to provide proper explanation on method section.
3) It would be better to add more clear procedure of Morris water maze test with some references along with proper explanation of escape latency.
4) In Figure legend of Figure 10, "n=?" number is missing.
5) In figure 11, size of scale bar is missing.
Author Response
Q1: In materials and methods, it is better to add the catalog number of every products or kits mentioned in the method section along with the company.
A1: Thanks for your suggestion, we have added it
Q2: In animal experiment, Is there any specific reason of using only Female transgenic mice rather than using both male and female mice to avoid the gender bias. The results may not be representative of the entire population and may not be applicable to male mice or humans. Better to provide proper explanation on method section.
A2: Constituting around 70% of individuals with AD, women have a greater lifetime risk for AD than men, and display approximately a threefold higher rate of disease progression with a broader spectrum of cognitive symptoms[22, 23]. Therefore, only female animals were used in this experiment.
[22]Xiong, J.; Kang, S. S.; Wang, Z.; Liu, X.; Kuo, T. C.; Korkmaz, F.; Padilla, A.; Miyashita, S.; Chan, P.; Zhang, Z., et al., FSH blockade improves cognition in mice with Alzheimer's disease. Nature 2022, 603, (7901), 470-476.
[23]Marongiu, R., Accelerated Ovarian Failure as a Unique Model to Study Peri-Menopause Influence on Alzheimer's Disease. Front Aging Neurosci 2019, 11, 242.
Q3: It would be better to add more clear procedure of Morris water maze test with some references along with proper explanation of escape latency.
A3: Thanks for your suggestion, we have added it
“The diameter of the circular pool is 120 cm, the height is 50 cm, the water depth in the pool is 30 cm, the bottom of the pool is white, and the water temperature is kept at (23±2)°C; the pool is divided into four quadrants, and a quadrant is selected in the center to place a platform at the same distance from the center of the pool wall. 12 cm in diameter, 29 cm in height, submerged 1 cm in water. In the experimental training stage, the mice were placed in the maze for training once a day and adapted to the environment. After 6 consecutive days, the learning and memory function tests were performed. An automatic camera and Xmaze analysis system were used to track and record animal movement trajectories in real-time.
The time of crossing the platform is an important indicator of the space search experiment. At a certain time, the number of times the experimental animals crossed the original platform. The more times the animal crossed the original platform, the better its spatial learning and memory ability[24].
Latency is an important indicator of the positioning and navigation stage of the Morris water maze, and it is the time required for the animal to successfully find the platform for the first time after each entry into the water. Its length also represents the quality of the animal’s spatial learning and memory ability, and the short incubation period indicates that the animal’s learning and memory ability is good[25]. The time in the quadrant where the platform is located in an index to evaluate the learning ability of animals. The longer the time and distance of animals in this quadrant, the better the spatial memory ability of animals[26].”
- Yang, L.; Wu, C.; Li, Y.; Dong, Y.; Wu, C. Y.; Lee, R. H.; Brann, D. W.; Lin, H. W.; Zhang, Q., Long-term exercise pre-training attenuates Alzheimer's disease-related pathology in a transgenic rat model of Alzheimer's disease. Geroscience 2022,44, (3), 1457-1477.
- Filip, T.; Mairinger, S.; Neddens, J.; Sauberer, M.; Flunkert, S.; Stanek, J.; Wanek, T.; Okamura, N.; Langer, O.; Hutter-Paier, B., et al., Characterization of an APP/tau rat model of Alzheimer's disease by positron emission tomography and immunofluorescent labeling. Alzheimers Res Ther 2021,13, (1), 175.
- Joo, I. L.; Lam, W. W.; Oakden, W.; Hill, M. E.; Koletar, M. M.; Morrone, C. D.; Stanisz, G. J.; McLaurin, J.; Stefanovic, B., Early alterations in brain glucose metabolism and vascular function in a transgenic rat model of Alzheimer's disease. Prog Neurobiol 2022,217, 102327.
Q4: In Figure legend of Figure 10, "n=?" number is missing.
A4: I am very sorry, due to our negligence, this information has been omitted, we have supplemented it in the article
Q5: In figure 11, size of scale bar is missing.
A5: Thanks for your suggestion, we have added it
Reviewer 2 Report
The authors submitted an interesting research article which deals with anti-inflammatory and neuroprotective ginsenosides activities. Since neurodegenerative disease, especially Alzheimer’s disease, shows increasing tendency of incidence, the topic of this manuscript is important.
The authors used adequate methods. The experimental design was well planned. Furthermore, the experimental data supported results and conclusions. Additionally, the authors presented in illustrative plots, tables and figures.
I found no serious errors. Therefore, I recommend the manuscript for publication.
Author Response
Thank you very much for your valuable time and your recognition of our work